# Effects of NADPH Oxidase Isoform-2 (NOX2) Inhibition on Behavioral Responses and Neuroinflammation in a Mouse Model of Neuropathic Pain

**DOI:** 10.3390/biomedicines11020416

**Published:** 2023-01-31

**Authors:** Luísa Teixeira-Santos, Eduardo Veríssimo, Sandra Martins, Teresa Sousa, António Albino-Teixeira, Dora Pinho

**Affiliations:** 1Departamento de Biomedicina—Unidade de Farmacologia e Terapêutica, Faculdade de Medicina, Universidade do Porto, 4200-319 Porto, Portugal; ats.luisa@gmail.com (L.T.-S.); eduverisousa@hotmail.com (E.V.); tsousa@med.up.pt (T.S.); albinote@med.up.pt (A.A.-T.); 2Centro de Investigação Farmacológica e Inovação Medicamentosa (MedInUP), Universidade do Porto, 4200-319 Porto, Portugal; 3Serviço de Patologia Clínica, Centro Hospitalar e Universitário São João (CHUSJ), 4200-319 Porto, Portugal; smvrmartins@gmail.com; 4EPIUnit, Instituto de Saúde Pública, Universidade do Porto, 4050-600 Porto, Portugal

**Keywords:** neuropathic pain, spared nerve injury, NOX2, GSK2795039, pain-related behavior, neuroinflammation, sex differences

## Abstract

NADPH oxidase isoform-2 (NOX2) has been implicated in the pathophysiology of neuropathic pain (NP), mostly through the modulation of neuroinflammation. Since it is also accepted that some neuroimmune mechanisms underlying NP are sex-dependent, we aimed to evaluate the effects of early systemic treatment with the NOX2-selective inhibitor (NOX2i) GSK2795039 on behavioral responses and spinal neuroinflammation in spared nerve injury (SNI)-induced NP in male and female mice. Mechanical sensitivity was evaluated with the von Frey test, while general well-being and anxiety-like behavior were assessed with burrowing and light/dark box tests. Spinal microglial activation and cytokines IL-1β, IL-6, and IL-10, as well as macrophage colony-stimulating factor (M-CSF) were evaluated by immunofluorescence and multiplex immunoassay, respectively. NOX2i treatment reduced SNI-induced mechanical hypersensitivity and early SNI-induced microglial activation in both sexes. SNI-females, but not males, showed a transient reduction in burrowing activity. NOX2i treatment did not improve their burrowing activity, but tendentially reduced their anxiety-like behavior. NOX2i marginally decreased IL-6 in females, and increased M-CSF in males. Our findings suggest that NOX2-selective inhibition may be a potential therapeutic strategy for NP in both male and female individuals, with particular interest in females due to its apparent favorable impact in anxiety-like behavior.

## 1. Introduction

Neuropathic pain (NP) pathophysiology involves a complex interplay of several mechanisms in the peripheral and central nervous system [1]. Reactive oxygen species (ROS) contribute to peripheral and central sensitization that underlies NP via a wide range of mechanisms, including the modulation of neuroinflammation [2,3]. A key feature of the nerve injury-induced neuroinflammatory response is microglial activation. Damaged afferent terminals release signaling molecules—such as macrophage colony-stimulating factor (M-CSF) [4]—that act as activators of microglia, which then proliferate within the spinal cord (SC) and undergo morphological changes [5,6]. A plethora of mediators, namely proinflammatory cytokines, M-CSF, and ROS [5,7], are synthetized and released, enhancing nociceptive transmission through the activation of ion channels, neuromodulation of excitatory and inhibitory synaptic transmission, and further induction of neuroinflammation and oxidative stress [2,6,8]. Anti-inflammatory cytokines and specialized pro-resolving mediators are also generated after nerve injury to promote the resolution of neuroinflammation and pain [8,9]. Some neuroimmune mechanisms that underlie NP are sex-dependent [10]. For example, microglia play a critical role in chronic pain processing in male rodents, while females preferentially use an alternative pathway which involves T-cells [11].

Restoring oxidative balance and concomitantly ameliorating neuroinflammation is a potential therapeutic strategy for NP [3]. However, general antioxidants have failed to demonstrate significant benefits in most clinical trials and some even induced detrimental effects [12], despite the antinociceptive effects that have been consistently demonstrated in experimental models [2]. Since ROS also have physiological functions, targeting specific disease-relevant ROS sources could be a better approach, instead of indiscriminately eliminating ROS once formed [13,14]. In this context, NADPH oxidases (NOXs), whose primary function is the regulated production of ROS [15], have emerged as potential pharmacological targets for NP, which is corroborated by mounting evidence implicating some NOX isoforms in NP development [3,16]. Specifically, NOX2 is upregulated in the nervous system after nerve injury [17,18,19,20,21,22], and NOX2-knockout mice exhibit attenuated NP behaviors [17,20,23,24]. Furthermore, NOX2-derived ROS are involved in spinal neuroimmune activation that underlies NP, namely microglia activation, and proinflammatory cytokine expression [17]. “Traditional” and peptide NOX inhibitors are either not specific/isoform-selective (e.g., apocynin; diphenyleneiodonium, DPI), or have inherent properties that limit their therapeutic utility (e.g., NOX2ds-tat, with potential to cause antigenic responses) [25,26].

Isoform selectivity is considered essential for a putative pharmacological agent with minimal off-target effects, and efforts to develop new small-molecule NOX inhibitors with those characteristics have increased in the last decade [26,27]. GSK2795039, a direct and CNS-permeable NOX2 inhibitor (NOX2i), was the first small molecule to demonstrate isoform 2-selectivity in vivo, acting through competition for the NADPH binding site of NOX2 [28]. To date, the effects of its systemic administration have been reported in rodent models of acute pancreatitis, lung injury, intracerebral hemorrhage, diabetes, liver fibrosis, and muscle physiology assays [28,29,30,31,32,33]. In traumatic brain injury models, GSK2795039 significantly downregulated NOX2 expression and activity in brain, attenuated neurological deficits and apoptosis, facilitated long-term potentiation, and decreased spontaneous synaptic transmission [34,35].

This study aimed to evaluate the effects of early systemic treatment with the NOX2i GSK2795039 on behavioral responses and neuroinflammatory parameters in male and female mice with spared nerve injury (SNI)-induced NP, and whether treatment outcomes are sex-specific.

## 2. Materials and Methods

### 2.1. Animals

Sixty-eight male (n = 34) and female (n = 34) C57BL/6J mice (specific pathogen-free; 8 weeks-old upon arrival) were obtained from Charles River (Lyon, France). Animals were housed in same-sex groups of two or four, under controlled temperature (21–24 °C) and humidity (45–55%) conditions, in a 12-h light/dark cycle (lights on at 8 AM). Food and water were provided ad libitum, and nesting material and cardboard tubes (one per cage) were used as environmental enrichment.

All experiments were performed in accordance with national (Portuguese Decree-Law 113/2013) and international (European Directive 2010/63/EU) guidelines for experimental research in animals, and specific guidelines for the study of pain (IASP Guidelines for the Use of Animals in Research, [36]). The study was approved by the local institutional Animal Welfare Committee (ORBEA-FMUP_57_2017/1812), and by the competent national authority, *Direção-Geral de Alimentação e Veterinária* (DGAV, ref. 0421/000/000/2020, 24/03/2020).

### 2.2. Experimental Design and Drug Administration

Upon arrival, animals were acclimatized for at least 1 week to the animal facility, and were then habituated to the experimenter and experimental settings for another 7–10 days before the beginning of the experimental procedures.

Sixty-four mice were randomly allocated, within sex, to one of two different experimental protocols that differed in euthanasia and tissue collection timepoints: “short,” tissue collection on day 2 or “long,” tissue collection on day 11 (the day of SNI surgery is counted as day 0). Then, they were randomly allocated within sex to treatment groups SNI-NOX2i or SNI-vehicle (Figure 1).

Based on a biologically relevant effect size-to-standard deviation ratio *ca.* 1.6 for the behavioral evaluation of pain, we estimated a sample size of 8 experimental units per group, using InVivoStat power analysis module, for a power of 80% and a significance level of 5%. Since each animal could be randomly allocated to either treated or vehicle group, even though animals were housed in groups (the drug vs. vehicle treatment was administered by s.c. injection), each animal constituted an experimental unit. Therefore, we divided the animals into 16 sets of 4 animals (1 vehicle-treated SNI male + 1 vehicle-treated SNI female + 1 NOX2i-treated SNI male + 1 NOX2i-treated SNI female). Eight sets were used for “short protocol” experiments and 8 for “long protocol.” Four of the 8 sets for each protocol were used to obtain fixed tissue (for immunofluorescence assays), whereas the other 4 were used to collect fresh tissue (for multiplex assays). An additional set of animals (part of a pilot study for a future work) was used for the burrowing test and euthanized at a later timepoint. A complete description of the number of animals initially allocated to the study, randomly assigned to each experimental group, used in each experimental procedure, and included in the analyses, with explanation for exclusion of animals/data points or other missing data, is provided in Appendix A.

According to the treatment group, a 10-mL/kg dose volume of the NOX2 inhibitor GSK2795039 solution (70 mg/kg, MedChemExpress), or of vehicle alone (10% DMSO, 40% PEG300, 5% Tween-80, and 45% saline) was subcutaneously injected into SNI mice twice daily, with the first dose administered 1 h before SNI surgery. This treatment scheme was based on previous data by Hirano et al. [28]: (i) complete NOX2 activity inhibition was obtained 2 h post-100 mg/kg intraperitoneal (i.p.) dose, returning to baseline levels 24 h post-dose; therefore, twice-daily injections should maintain a sustained NOX2 inhibition during the first 48 h after SNI; (ii) 100 mg/kg GSK2795039 was well tolerated when subcutaneously administered twice-daily during 5 days; (iii) GSK2795039 dose-dependently inhibited ROS production, achieving *ca.*50% and 95% inhibition at 2 and 100 mg/kg, respectively; therefore, 70 mg/kg significantly inhibits ROS without complete abrogation.

Pain-related behavior was evaluated during the light cycle phase in mice allocated to the “long protocol,” with von Frey and burrowing tests (baseline, and days 4 and 9), and light/dark box (LDB) test (day 10). Tissue collection timepoints were chosen: to allow an evaluation of SNI-induced changes at the onset phase, and the acute effects of the NOX2i treatment (less than 24 h after the last administration of NOX2i)—day 2; to assess whether an early short-term NOX2 inhibition could result in sustained changes that were still observable several days after NOX2i treatment completion—day 11.

### 2.3. Neuropathic Pain Induction

NP was induced in 12-week-old mice by the SNI model, according to the method of Bourquin et al. [37], under general anesthesia with isoflurane (5% for induction and 2–2.5% for maintenance). Briefly, after skin incision at mid-thigh level of the left hindlimb, a section was made through the *biceps femoris* muscle, exposing the sciatic nerve and its three peripheral branches. The common peroneal and tibial nerves were tightly ligated together using a 6.0 silk thread (Fine Science Tools), and transected distal to the ligation, removing 1–2 mm of the nerve stump. The sural nerve was preserved by avoiding nerve stretching or contact with surgical instruments. Muscle and skin were closed in two distinct layers with silk 5.0 suture (Silkam, B. Braun Medical). The surgical site was examined in each animal after euthanasia, to confirm that the sural nerve was intact and no nerve regeneration had occurred. A damaged sural nerve was detected in a vehicle-treated SNI-female from the “long protocol,” which was therefore excluded from the study.

Sham-operated animals were submitted to a similar procedure, but without nerve ligation and transection. Instead, after exposure of the sciatic nerve and its three peripheral branches, a 2–3-mm-long 6.0 silk thread was placed at the level of the trifurcation, before muscle and skin closure.

### 2.4. Behavioral Assessment

Behavioral assessments were conducted only on “long protocol” mice. Baseline evaluations of burrowing (SNI-veh, 9 male and 7 female; SNI-NOX2i, 9 male and 9 female) and mechanical allodynia (SNI-veh, 7 male and 6 female; SNI-NOX2i, 7 male and 7 female) were performed prior to SNI induction (see Figure 1). Burrowing was assessed during the morning hours, and the von Frey test was performed at least 3 h later. Tests were repeated on the same animals at 4 and 9 days after SNI induction. The LDB test was conducted only at day 10 after SNI (SNI-veh, 7 male and 6 female; SNI-NOX2i, 7 male and 7 female). Exclusions and attritions are reported and justified in Appendix A.

Prior to each test, animals were allowed to acclimatize to the testing room for at least 1 h. Behavioral testing was performed by a female experimenter blinded to the treatment group during data collection.

#### 2.4.1. Burrowing Test

The burrowing test, based on a spontaneous behavior of rodents to remove items from an enclosed tube, is often used as a surrogate measure of well-being in mice, namely in pain models [38,39].

An adapted version of the protocol described by Shepherd et al. [40] was conducted. Mice were acclimatized to the burrowing tube on the day before the burrowing assay (for 1 h) and on the testing day (for 30 min) by placing the empty tube into their home cage, so that all animals were simultaneously exposed to it. Exposing mice to the burrowing tube in groups enables social facilitation of their exploratory behavior [41]. The burrowing tubes consisted of red-colored acrylic cylinders (inner diameter: 7.5 cm, length: 15 cm), with one end sealed. On testing day, tubes were filled with 180 g of the corncob bedding used in the home cage, and carefully placed in one corner of the testing cage. Approximately 10 g of bedding from the home cage was transferred to the testing cage immediately before the test, to minimize the new environment-related stress and potentially different olfactory cues. Each mouse was tested individually by being placed for 15 min in its respective testing cage. In the end, the bedding material remaining in the tube was weighed. Tubes were rinsed with 35% ethanol between animals. Each testing cage and its respective bedding were used for the same animal throughout the entire protocol.

In our experiments, some animals did not show interest or were reluctant to voluntarily enter the tube during the acclimatization period in the home cage. When individually placed in the testing cage for the first time, those animals did not interact with the bedding material inside the tube. Since a practice run has been proposed to improve burrowing performance and reduce variability between animals [41], three testing days were conducted before SNI surgery. Baseline measurements presented herein correspond to the results from the third day. One female SNI-vehicle mouse did not burrow at baseline and, for that reason, was excluded from the analysis.

The burrowing activity, calculated by subtracting the weight of bedding remaining in the tube at the end of the experiment from the starting weight, was expressed as the proportion (%) of displaced bedding [40].

#### 2.4.2. Von Frey Test

Mechanical withdrawal threshold (MWT) was assessed with the von Frey test, using the “ascending stimulus” method [42]. Mice were placed in transparent acrylic cylinders (inner diameter: 8.4 cm; height: 7 cm), wrapped in a grey plastic strip to darken the inside, on a 5 × 5 mm square metal grid floor (Ugo Basile). Animals were habituated to the apparatus for 30 min during at least three days before the first test, and, on each test day, for 10 min before beginning the procedure. A series of 9 calibrated von Frey filaments (0.008, 0.02, 0.04, 0.07, 0.16, 0.4, 0.6, 1, and 1.4 g; Stoelting Co., Wood Dale, IL, USA) was used. Filaments were perpendicularly applied to the sural nerve skin territory of the plantar surface of the ipsilateral hind paw until bending. Five stimuli were applied with each filament over a total period of 30 s (approximately 2 s per stimulus), starting with the lowest force and proceeding in ascending order. MWT was defined as the monofilament that first evoked 3 out of 5 positive responses (sudden paw lifting, guarding, licking, or flinching) [43].

#### 2.4.3. Light/Dark Box Test

The LDB test, an unconditioned test based on an approach-avoidance conflict, aims to assess rodents’ anxiety-like behavior by exploring the conflict between the tendency of mice to spontaneously explore new environments and their natural aversion to brightly-illuminated areas [44,45].

The LDB apparatus consisted of an acrylic testing box with two compartments (one transparent and one opaque black—15 × 15 × 20 cm each), interconnected through a 5 × 5 cm hole to allow free movement between the two areas. The transparent chamber was further illuminated by placing a bulb (330 lm) above it, while the dark chamber was covered with a black lid. At the beginning of the session, each mouse was released in the light compartment with its head facing away from the door and allowed to explore the arena for 5 min. Test sessions were recorded using a video camera (Sony HandyCam HDR-CX240E). The box was cleaned with 35% ethanol between subjects to eliminate olfactory cues.

Behaviors analyzed in the LDB test included the time spent in the light chamber (%), and two exploratory behaviors: the number of full-body transitions between chambers, and the number of rearing events in the light chamber, standardized by the time spent in that chamber (number of rearing events/time spent in the light compartment). A rearing event is defined to occur when mice put their weight on their hind legs, raising both forelimbs from the ground, and extending the head upwards. These parameters were analyzed using Solomon Coder Software (Version beta 17.03.22; https://solomon.andraspeter.com/, accessed on 13 June 2019). Decreased time spent in the illuminated compartment, decreased number of transitions between chambers, and decreased number of rearing events have been considered indicators of anxious-like behavior in this apparatus [44,46].

### 2.5. Tissue Collection

On days 2 (for “short protocol” mice) and 11 (for “long protocol” mice), fresh (SNI-veh, 4 male and 4 female per protocol; SNI-NOX2i, 4 male and 4 female per protocol) and fixed (SNI-veh, 4 male and 4 female per protocol; SNI-NOX2i, 4 male and 4 female per protocol) tissues were collected.

Mice were deeply anesthetized with i.p. sodium pentobarbital (100 mg/kg; Euthanimal, NePhar, Mem Martins, Portugal), and transcardially perfused through the ascending aorta with phosphate-buffered saline (PBS) 0.1 M pH 7.4 only (fresh samples), or PBS followed by 4% paraformaldehyde (fixed samples). SC L3-L5 segments were collected, and either snap-frozen in liquid nitrogen and stored at −80 °C (fresh samples), or post-fixed in the same fixative for 3 h and then transferred to a 30% sucrose solution with 0.1% sodium azide, at 4 °C (fixed samples).

### 2.6. Immunofluorescence

Fixed SC sections were sliced using a cryostat (Leica CM3050) into 30 μm transverse sections (4 series). The contralateral side was identified with a small cut in the ventral horn. Free-floating sections were stored at −20 °C in a cryoprotectant solution of 30% (*m/v*) sucrose dissolved in phosphate buffer 0.1 M and 30% (*v/v*) ethylene glycol, until analysis.

Immunofluorescence staining for ionized calcium binding adaptor molecule 1 (Iba-1) was performed to assess microglial activation. Briefly, one series of SC sections of each animal was washed in 0.1 M PBS, treated with 1% sodium borohydride in PBS for 30 min, and further washed in PBS and in PBS with 0.3% of Triton X-100 (PBST). Then, the sections were incubated for 2 h with a blocking solution containing 0.1 M glycine and 10% normal horse serum (NHS, Gibco, Cat. No. 16050130, Auckland, New Zealand) in PBST to minimize background staining, and incubated for 2 overnights at 4 °C with a rabbit anti-Iba-1 primary antibody (1:1000; Fujifilm Wako, Cat. No. 019-19741, Neuss, Germany) diluted in PBST with 2% NHS. Sections were subsequently washed in PBST and incubated for 1 h with Alexa Fluor-594 donkey anti-rabbit secondary antibody (1:1000; Invitrogen, Cat. No. A21207, Waltham, MA, USA). Finally, after repeated washing with PBST and PBS, sections were mounted on gelatin-coated slides and coverslipped with Prolong Gold antifade mounting reagent with DAPI (Invitrogen Ltd., Cat. No. P36941). Negative controls were performed by omitting the incubation with the primary antibody.

Images of the immunofluorescence reactions were acquired with 2.5 × and 10 × objectives, using a fluorescence microscope (Axio Imager.Z1, Zeiss), through an AxioCam MRm digital camera with AxioVision 4.8.2 software (Carl Zeiss MicroImaging GmbH, Oberkochen, Germany), under the same image acquisition settings. Iba-1 fluorescent intensity was determined in images captured with a 10 × objective, using the open-source software Fiji [47] in a blinded manner. Images were first converted to 8-bit grayscale and mean grey values were automatically measured within a rectangular region of fixed dimensions (266 × 159 µM) comprising the medial two-thirds of the dorsal horn (laminae I-III) [48]. Both ipsilateral and contralateral sides were evaluated, and the average ipsilateral/contralateral ratio was calculated to normalize the data. Sections severely damaged in the region of interest were not analyzed and only animals with a minimum of 5 sections available were included in the statistical analysis.

### 2.7. Multiplex Assay

Frozen samples were homogenized in cold lysis buffer (300 µL per 10 mg of tissue) composed by 50 mM Tris-HCl pH 7.4, 150 mM NaCl, 1% IGEPAL, and protease and phosphatase inhibitors (cOmplete, Mini Protease Inhibitor Cocktail and PhosSTOP Phosphatase Inhibitor Cocktail, Roche). Tissues were sonicated in a bath sonicator at 4 °C for 3 × 5 min. After centrifugation (12,000× *g*, 4 °C, 10 min), supernatants were aliquoted and stored at −80 °C until multiplex assay was performed. Total protein concentration was quantified by the Bradford method using bovine serum albumin (BSA) as a standard.

IL-1β, IL-6, IL-10, and M-CSF were simultaneously quantified in each sample using Milliplex MAP Mouse Cytokine/Chemokine Magnetic Bead Panel (Merck Millipore, Burlington, MA, USA) on a Luminex 200 analyzer (Millipore), according to the manufacturer’s protocol. Before the assay, the protein concentration of each sample was adjusted to 2 mg/mL with lysis buffer. This immunoassay uses dual-laser flow cytometry-based technology, and involves incubating the protein extract with fluorescent-coded magnetic beads pre-coated with capture antibodies, followed by sequential incubation with biotinylated detection antibodies and streptavidin-phycoerythrin conjugate [49]. Results are presented as picograms-per-mg of total protein.

### 2.8. Statistical Analyses

Statistical analyses were performed with InVivoStat 4.3.0 (Cambridge, UK), a statistical software package that uses R as its statistics engine [50]. Graphic illustrations were created with GraphPad Prism 9.2.0 for Windows (San Diego, CA, USA). Results are expressed as predicted means (95% confidence interval, CI) unless otherwise stated. The significance level was set at 5%. Assumptions for the use of parametric analysis were ascertained using diagnostic plots: *Normal probability plot* for the normal distribution of the residuals’ assumption, and *Predicted* vs. *Residuals plot* for the homogeneity of the variance assumption. Log10 transformation was applied to von Frey MWT, since the original values fulfilled neither of the required assumptions (see also [51]).

Results from LDB test, Iba-1 immunofluorescence, and multiplex quantifications were analyzed with single measures parametric analysis (ANOVA), with factors *Treatment* (NOX2i, vehicle) and *Sex* (male, female). Planned comparisons between groups were performed with the least significant difference (LSD) test and corrected with Holm’s procedure for multiple comparisons. Log10-transformed MWT and % of burrowing activity were analyzed with repeated measures parametric analysis (mixed model), with *Treatment* and *Sex* factors, and with *Time* (baseline, day 4, day 9) as a repeated factor. Planned comparisons were further conducted to assess differences both between and within groups, with Holm’s correction for multiple comparisons. In both single and repeated measures procedures, when no *Sex* main effects or interactions were detected, data from males and females within the same treatment group were pooled [52,53] (see Appendix A). Correlation analyses were performed with the Pearson coefficient.

For practical reasons, in vivo procedures were divided into several mini-experiments, or blocks, such that each block contained an integer multiple of the 4-mice set described in Section 2.2, and statistical analysis accounted for the blocking factor whenever a blocking effect was detected (see Appendix A).

## 3. Results

### 3.1. NOX2i Treatment Did Not Increase the Well-Being of SNI-Mice

In the burrowing test, the mixed model analysis revealed a main effect of sex (*F*(1, 30) = 11.41, *p =* 0.0020), with males burrowing more extensively than females, independently of the treatment group or the testing day, and time (*F*(2, 60) = 12.81, *p* < 0.0001) (Figure 2). The burrowing activity decreased after surgery in most male and female SNI animals on day 4 (although this reduction was only statistically significant in females; ♀-SNI-veh: BL vs. day 4, *p =* 0.036; ♀-SNI-NOX2i: BL vs. day 4, *p* = 0.048), but not on day 9 (Figure 2). Therefore, SNI surgery affected, at short term, the motivation of mice (particularly females) to engage in a voluntary activity not essential in the laboratory environment.

No differences were found on the effects of NOX2i treatment (*F*(1, 30) = 0.07, *p =* 0.79; Figure 2). Thus, NOX2i treatment did not significantly improve the reduced well-being of mice on day 4.

### 3.2. NOX2i Treatment Reduced SNI-Induced Mechanical Hypersensitivity in Both Male and Female Mice

To assess whether NOX2i treatment influenced SNI-induced mechanical hypersensitivity, we evaluated the MWT of the ipsilateral paw before surgery and on days 4 and 9. Since no differences between sexes were detected, the final analysis did not include *Sex* factor (as described in Section 2.8; Appendix A). Although the mixed model analysis revealed no main effects of treatment (*F*(1, 21) = 1.49, *p =* 0.24), both time (*F*(2, 50) = 217.7, *p* < 0.0001) and treatment by time interaction (*F*(2, 50) = 6.42, *p =* 0.0033) were statistically significant. No significant differences were observed between groups at baseline (SNI-NOX2i vs. SNI-veh, *p =* 0.12). After surgery, SNI animals developed mechanical hypersensitivity, as confirmed by the decrease in MWT on days 4 and 9, compared to baseline (SNI-veh: BL vs. day 4, *p =* 0.0004; SNI-veh: BL vs. day 9, *p =* 0.0004; SNI-NOX2i: BL vs. day 4, *p =* 0.0004; SNI-NOX2i: BL *vs.* day 9, *p =* 0.0004; Figure 3). When compared to vehicle-treated mice, treatment with NOX2i increased MWT by 14.3% on day 4 (*p =* 0.038, NOX2i-treated vs. vehicle-treated mice; Figure 3), and by 14.4% on day 9 (*p =* 0.040, NOX2i-treated vs. vehicle-treated mice; Figure 3).

### 3.3. NOX2i Treatment Appeared to Improve Anxiety-Like Behavior on SNI Female Mice

We used the LDB test to assess anxiety-like behavior. Many different outcomes have been evaluated in this test [44]. However, it is now generally accepted that simultaneous evaluation of several parameters is the best method to provide a global analysis of mice behavior, thus enabling a better interpretation of the anxiety level of each animal (see, e.g., [45]). Therefore, we assessed three parameters, considered as indexes of aversion and/or exploratory activity: increased time spent in the anxiogenic zone (the illuminated compartment), multiple transitions between chambers, and increased rearing behaviors in the light chamber. All of these parameters are suggested to reflect low anxiety-like behavior [44,45]. Since motor function has been described to remain unaffected in SNI-mice [54,55,56], we do not expect the tests based on exploratory-driven paradigms to be affected by limitations on locomotion brought about by our NP model.

C57BL/6 mice have been characterized as highly active with moderate levels of anxiety [45,57], and “intermediate-reactive” in the LDB test [58]. Such characteristics should, at least theoretically, provide a large “behavioral window” (with ability to decrease/increase their behavior), which is important to enable the detection of potential effects [45]. In our study, a high variability in mice behavior was observed, thus hampering conclusions about the impact of NOX2i treatment on day 10. In addition, some vehicle-treated animals spent more than 50% of the time in the light area, which suggests that the illuminated side did not induce a clear aversion in some SNI-mice [45]. Correlation analysis of our results showed that the number of transitions between the two compartments and the number of rearing events, two exploratory-driven behaviors, but not the time spent in the light, were significantly correlated (number of transitions between chambers *vs*. number of rearing events: correlation coefficient, 0.596; test statistic, 3.708; *p =* 0.0010; other correlations: *p* > 0.05; Figure 4D).

The administration of NOX2i did not significantly affect the time spent in the light chamber or the number of transitions between chambers (largest *F* = 2.34, *p =* 0.14, see Appendix A; Figure 4A,B). However, the number of rearing behaviors per unit time spent in the light chamber was differently affected by NOX2i treatment in male and female mice (*F*(1, 23) = 5.73, *p =* 0.0252 for treatment by sex interaction; Figure 4C), and NOX2i-treated females displayed a higher number of rearing behaviors in light than their vehicle-treated controls, although this difference was not statistically significant after the Holm’s correction for multiple comparisons (*p =* 0.15). Likewise, a tendency was apparent for a higher number of full body transitions between compartments in NOX2i-treated females (Figure 4B), as compared to their vehicle-treated controls (again, not statistically significant). Therefore, both parameters suggest some anxiolytic activity of NOX2i in females, but not in males.

### 3.4. NOX2i Treatment Reduced Early SNI-Induced Spinal Microglial Activation in Both Male and Female Mice

To investigate putative mechanisms underlying the NOX2i effects, the immunoreactivity of Iba-1 was evaluated on the SC dorsal horns collected on days 2 and 11. Although the fact that Iba-1 is a pan-myeloid marker should be acknowledged, Iba-1 is still a reliable microglial marker and has been the most widely-used marker for microglial activation in NP models [59,60].

Previous studies have reported that microglial activation following peripheral nerve injury begins to increase three days after injury and peaks around postinjury-days 7–14 [61,62]. Our results show a similar temporal pattern, as the SNI-induced Iba-1 immunoreactivity increase in the spinal dorsal horn was more marked on day 11 than on day 2 (Figure 5C,D).

Since no differences between sexes were detected, the final analysis did not include the *Sex* factor (as described in Section 2.8; Appendix A). NOX2i treatment decreased Iba-1 fluorescence intensity in the ipsilateral dorsal horn by 9.0% on day 2 (*F*(1, 13) = 13.05, *p =* 0.0032; Figure 5A,C) but the effect was not detected on day 11 (*F*(1, 13) = 0.57, *p =* 0.46; Figure 5B,D).

Although not quantified herein, since the focus of our study is nociception, a remarkable increase in Iba-1 immunoreactivity in the ipsilateral ventral horn (surrounding the motor neurons somata) was also observed in SNI-animals, as previously described [62,63,64], particularly on day 11 (Figure 5B).

### 3.5. NOX2i Treatment Marginally Decreased IL-6 in SNI Female and Increased M-CSF in SNI Male Mice on Day 11

We also investigated the possible modulation of spinal pro- and anti-inflammatory cytokines in SNI-mice treated with vehicle and NOX2i.

Two days after surgery, no statistically-significant differences between NOX2i- and vehicle-treated male and female mice were found in any of the analyzed cytokines (largest *F* = 3.39, *p =* 0.090, for the *Treatment* factor in M-CSF, see Appendix A; Figure 6A,C,E,G).

On day 11, no differences between sexes or treatment groups were found on either proinflammatory IL-1β (largest *F* = 0.15, *p =* 0.71, see Appendix A; Figure 6B) or anti-inflammatory IL-10 (largest *F* = 1.06, *p =* 0.32, see Appendix A; Figure 6F) concentrations. On the other hand, NOX2i treatment differently affected the proinflammatory IL-6 spinal concentration in males and females (Treatment by Sex interaction: *F*(1, 12) = 8.36, *p =* 0.014; Figure 6D). In females, NOX2i appeared to have a beneficial effect, since IL-6 values were 21.5% lower in NOX2i- than in vehicle-treated individuals (borderline *p*-value of 0.060), while no significant difference was found in males. M-CSF concentration was also significantly influenced by NOX2i treatment (*Treatment* factor: *F*(1, 12) = 6.37, *p =* 0.027) on day 11. Specifically, NOX2i treatment increased M-CSF values in males by 14.8% (*p =* 0.030) (Figure 6H).

In addition, a global correlation analysis showed that M-CSF was significantly positively correlated with IL-1β on day 2 (correlation coefficient: 0.572; *test statistic*: 2.608; *p =* 0.021; Figure 7A), and with IL-6 on day 11 (correlation coefficient: 0.510; *test statistic*: 2.217; *p =* 0.044; Figure 7B). This latter correlation was still found when restricting the correlation analysis to either NOX2i-treated mice (correlation coefficient: 0.788; *test statistic*: 3.134; *p =* 0.020) or vehicle-treated mice (correlation coefficient = 0.823; *test statistic* = 3.550; *p =* 0.012). Furthermore, on the same day, an inverse correlation between IL-1β and IL-10 was found in the SNI-vehicle male subgroup (correlation coefficient = −0.979; *test statistic* = −6.788; *p =* 0.021).

## 4. Discussion

Current knowledge about the role of NOX2 in NP has primarily arisen from studies with NOX2-knockout mice. However, full validation of NOX2 as a drug target also requires evaluation of the effects of its inhibition by pharmacological agents [26], which has been hampered by the limited number of isoform-selective options available. Indeed, to date, only the peptide NOX2ds-tat had been evaluated in this context, and even those studies are scarce. Therefore, to the best of our knowledge, the present study is the first to report the effects of a treatment with a small-molecule NOX2-selective inhibitor on experimental NP. Our results demonstrate that short-term early systemic treatment with GSK2795039 attenuates SNI-induced mechanical hypersensitivity and microglial activation at early timepoints. Furthermore, although no impact on mice well-being was found, NOX2i treatment appears to have other beneficial effects on SNI-females, since it marginally improved SNI-induced anxiety-like behavior and decreased spinal proinflammatory IL-6 on day 11. Conversely, M-CSF values were increased in NOX2i-treated males on the same day, which agrees with the lack of therapeutic effect on microglial activation at this timepoint.

### 4.1. On the Behavioral Assessment of Analgesic and Anxiolytic Effects of NOX2i Treatment

NP-induced mechanical and thermal hypersensitivity are reduced in NOX2-deficient mice [17,24]. The systemic administration of NOX2ds-tat once daily for 7 days, starting 3 h after spinal cord injury, yielded antinociceptive effects at postinjury days 27, 34, and 55 [23], whereas no immediate effect was found on mechanical hypersensitivity after a single intraperitoneal injection of NOX2ds-tat, 10 days after partial sciatic nerve ligation (PSNL) [65]. In our treatment scheme, we administered the first dose shortly before surgery, aiming to inhibit NOX2 during the first 48 h after SNI. The attenuated mechanical hypersensitivity we observed on days 4 and 9 confirms the influence of the timing of NOX2-inhibition on the onset of hypersensitiveness, and further suggests that ROS are more involved in its onset than in its maintenance.

Since ongoing pain is the predominant complaint of chronic pain patients [66], and NP is often associated with co-morbidities such as anxiety and depression [1], a comprehensive assessment of an animal’s pain experience that better compares with the human condition should not be exclusively based on stimulus-evoked reflexive responses [67,68,69]. Moreover, oxidative stress has been suggested to be a key contributor to anxiety states [70,71], and the inhibition of NOX enzymes has been associated with anxiolytic-like effects [72,73]. Accordingly, we performed behavioral tests to assess well-being, which may provide insight into how an animal’s affective state is affected by ongoing pain and anxiety-like behaviors.

Although the burrowing test may be considered a general welfare evaluator, there are numerous reports of burrowing deficits that are reverted by analgesics in several NP models [40,74,75,76,77,78]. Our results support the hypothesis that burrowing activity is a pain-depressed behavior affected by SNI, especially at early postsurgical timepoints, particularly in females. According to the literature, the SNI impact on the burrowing activity may depend on experimental conditions or even strain differences [38,79]. Using different mouse strains (CD-1 vs. C57BL/6) and methodologies (burrows filled with normal diet food pellets *vs*. corncob bedding), Bravo-Caparrós et al. reported that SNI did not alter burrowing behavior in female mice seven days after surgery [80], while Shepherd et al. found deficits in postsurgical days 5–25, and no significant differences between sexes in burrowing performance [40]. Using the same strain, and following a similar protocol as Shepherd et al., we found a significant overall difference between sexes, which we tentatively attribute to slight differences in experimental conditions, such as different tube volumes, which imply different initial amounts of bedding material.

According to our LDB test results, NOX2i-treated SNI females seemed to counteract the anxiety-like behavior of vehicle-treated SNI-females, as evaluated by the number of transitions between light and dark compartments, and the number of rearing events in light. Unlike what is described for most murine models [81], in C57BL/6J mice (our animal model), females have already been reported to exhibit more anxiety-like behaviors than males [82]. On the other hand, locomotion is commonly reported to be higher in females [83,84]. Since, in our study, vehicle-treated SNI-females tended to display a lower number of transitions and rearing behaviors than their male counterparts, we may infer that the behavior of NOX2i-treated females reflects, indeed, lower levels of anxiety. However, since no differences in the time spent in light were found, and we did not directly evaluate locomotion, definitive conclusions cannot be drawn. Conflicting results have been obtained by other groups studying the development of anxiety-like behavior following SNI [85,86,87,88,89], which may be associated with methodological variables such as test timepoint, housing conditions, light cycle phase in which testing occurs, manipulations preceding the test, combination/sequence of tests, or species/strain [56,88,90]. Sieberger et al. reported that, in the LDB test, both male and female C57BL/6 mice exhibited significant reduction in time spent in light between four and six weeks post-SNI, which was not attributable to reduction in locomotion [91]. Therefore, the assessment of mice behavior in the LDB test at later timepoints might be useful to confirm the anxiolytic-like effect of NOX2i. 

Ultimately, our behavioral results reflect the fact that spontaneous pain in the SNI model is acknowledged as difficult to measure [56], and that emotional-like behaviors have an intrinsic high variability [77,88].

### 4.2. On the Neuroinflammatory Mechanisms Underlying NOX2i Actions

Mechanistically, we focused on neuroinflammatory parameters in the SC. NOX2 inhibition during the early stages of NP development reduced SNI-induced microglial activation on day 2, which may, at least partly, account for the attenuation of mechanical hypersensitivity in NOX2i-treated mice. Our findings are in accordance with those by Kim et al., who reported that spinal nerve transection (SNT)-induced microglia activation was attenuated in NOX2-deficient mice on postinjury day 3 [17]. Pharmacological inhibition of microglia activation and proliferation has consistently been reported to attenuate NP behaviors in males [92], but not females [11,93,94]. Interestingly, we did not find significant differences between sexes in mechanical hypersensitivity or microglial activation. Priming effects may add a further level of complexity within the microglial involvement in NP, since microglia can adapt their response after experiencing an initial stimulus, reflecting a status of immune memory, which can either enhance (“immune training”) or suppress (“immune tolerance”) microglial responses to a subsequent insult, even after a long latent phase [95]. Microglial priming is likely influenced by exposure to immune challenges, stressors, and injuries [96,97], whose impact depends on several factors, such as age, type of stimuli, or the duration of exposure [98,99].

In microglia, NOXs are the main source of ROS, which, in turn, modulate the microglial generation of inflammatory mediators [3,100]. Indeed, there is a reciprocal relationship between NOX2-induced ROS production and microglial activation that underlies chronic pain, thereby propagating oxidative stress and neuroinflammation [3]. Redox signaling has a profound impact on transcription factors that modulate microglial fate, such as nuclear factor kappa-light-chain-enhancer of activated B cells (NF-κB), and nuclear factor (erythroid-derived 2)-like 2 (Nrf2), which are master regulators of the proinflammatory and antioxidant responses of microglia, respectively [100]. NOX2 has been shown to be involved in the activation of mitogen-activated protein kinases (MAPK) and phosphoinositide 3-kinase (PI3k), as well as in the regulation of NLRP3 inflammasome activity. Of note, PI3K, which has also been identified as an essential mediator of the memory-like effects of microglia [99], controls microglial ROS generation and has an important role in the regulation of NOX in microglia and microglial proliferation [101,102].

Among several pain-relevant mediators released by activated microglia, pro- and anti-inflammatory cytokines stand out as powerful neuromodulators [8] whose activation follows a tissue- and time-dependent pattern [8,103,104]. In SC, proinflammatory cytokines IL-1β and IL-6 have been generally reported to be increased after peripheral nerve injury [103,105]. IL-6 is currently considered a sex-independent mediator in pain hypersensitivity [10]; however, our results raise the possibility of a sex-specific role of IL-6, at least in the SNI model. Interestingly, results from Xu et al. show increased autotomy behavior in IL-6-deficient female, but not male, mice, following sciatic nerve transection [106], which also supports a female-specific effect of IL-6 after peripheral nerve injury. Several groups have reported that the intrathecal injection of IL-6 blocking antibodies or antagonists results in antinociceptive effects [105] but most of these studies were performed in male animals, and data from females are still lacking. Hence, whether the spinal IL-6 reduction in NOX2i-treated females found in our study is a beneficial effect of NOX2i treatment remains to be ascertained. It is noteworthy that Kim et al. reported that IL-1β gene expression was significantly reduced in NOX2-deficient mice with SNT-induced NP on postinjury day 3 [17]. However, we did not detect any effect of NOX2i treatment on IL-1β protein values.

The role of IL-10 in the sequence of events following nerve injury is still highly debated, and conflicting results have been described in studies examining IL-10 protein values within the SC after peripheral nerve injury [107,108,109,110,111,112,113]. The actions of IL-10 in NP involve the suppression of proinflammatory cytokines, including IL-1β [103]. Accordingly, on day 11, we found an inverse correlation between IL-1β and IL-10, specifically in the SNI-vehicle male subgroup.

Increased M-CSF mRNA levels have been reported in SNI-rat SC between days 2 and 14 [114]. ROS regulate M-CSF signaling, namely PI3K/Akt activation. In particular, NOX-induced ROS are required for the activation M-CSF receptor, Fms tyrosine kinase, and were shown to regulate M-CSF-induced monocyte/macrophage proliferation [115,116]. Furthermore, the application of DPI inhibited the responses of bone marrow monocyte/macrophage lineage cells to M-CSF and reduced M-CSF-induced cell proliferation [116]. Therefore, although we did not find differences in M-CSF protein values on day 2, the inhibition of NOX might have beneficially influenced its downstream pathways, which may explain the reduction in microglia activation in NOX2i-treated mice. On day 11, M-CSF values were higher in NOX2i-treated males than in their vehicle-treated controls. Indeed, between days 2 and 11, M-CSF decreased in the latter subgroup, while the former remained unaltered. Although further work is necessary to confirm this observation and elucidate the underlying mechanism, we speculate that NOX2i treatment might have triggered a counterregulatory/compensatory mechanism or a rebound effect. Moreover, increased M-CSF values in NOX2i-treated males may, at least partly, explain the lack of differences in microglia activation between experimental groups on day 11. Pain-inducing effects of M-CSF have been described as male-specific, since its administration is sufficient to induce mechanical hypersensitivity in male, but not female, mice [117,118]. Indeed, in female mice, regulatory T-cells seem to have a suppressive effect on the M-CSF-mediated immune activation [117]. Transcriptomic profiling and morphological analysis of dorsal horn microglia demonstrated that this effect correlated with robust microglial activation only in males [117], while another study reported similar microglial activation in both sexes, as assessed by Iba-1 immunoreactivity [118]. Likewise, we did not find significant differences in microglial activation between male and female mice.

In our study, M-CSF seemed to associate with exacerbated proinflammatory responses, since it positively correlated with proinflammatory cytokines at both timepoints. Specifically on day 11, positive correlations with IL-6 were further found when restricting the analysis to either NOX2i- or vehicle-treated groups. Curiously, IL-6 has also been implicated in the recruitment and activation of microglia after nerve injury [119], and M-CSF dose-dependently increased the expression of microglial IL-6 mRNA in vitro [120]. Although M-CSF has also been shown to induce IL-β gene expression in primary cultured microglia, no associated upregulation of proinflammatory mediators, namely IL-1β and NOX2, has been found in vivo [118].

### 4.3. On the Limitations of the Study and Future Research Perspectives

Although we focused on the inflammatory events in the SC, NOX2 contributes to NP pathophysiology not only by central, but also by peripheral, mechanisms [16]. Nerve injury induces NOX2 upregulation in the injured nerve [53] and dorsal root ganglia (DRGs) [20,22]. Conversely, Geis et al. found very mild NOX2 mRNA expression in the DRG and SC of CCI-mice, although the authors also reported strong NOX2 upregulation in the sciatic nerve on days 4 and 8 [21]. Nevertheless, NOX2 was shown to promote pain-relevant macrophage-neuron signaling in DRG [20], and the local application of NOX2ds-tat onto DRG 2 h before (but not 5 days after) surgery attenuated the SNI-induced mechanical hypersensitivity and hyperexcitability of DRG neurons [22]. Since we used a systemic route of administration, drug actions on the periphery might have contributed to the antinociceptive effect of NOX2i, and further studies are necessary to assess this possibility. On the other hand, potential side-effects of GSK2795039 due to its weak activity against other flavoproteins and NOX isoforms [121] cannot be excluded, and other putative mechanisms of action, e.g., antioxidative or antiapoptotic, were not investigated. Future work will evaluate the effects of NOX2i treatment on ROS production, and on NF-κB and MAPK pathways.

Another possible limitation of this study is the exclusive use of young adult mice. Traumatic peripheral nerve injuries are known to result in different consequences according to rodents’ age (e.g., nerve injury does not induce mechanical allodynia in young individuals [122,123], which is explained, at least partly, by absent/weak microglial activation and increased expression of anti-inflammatory mediators [123,124,125,126]). Indeed, the pattern of adaptative responses in microglia also depends on the developmental state of these cells [127]. Unstimulated newborn naïve microglia have showed lower levels of ROS in vitro than adult and aged microglia [127], and prior experiments in rodents with spinal cord injury (SCI) have demonstrated increasing NOX2 expression with age [128,129,130]. Using *post-mortem* midbrain tissues of young and elderly adults without diagnosed neurodegenerative diseases, Geng et al. found increased ROS production, and upregulated microglial density and NOX2 expression, along with IL-1β production, in the aged human tissue [131]. Therefore, the therapeutic efficacy of ROS-based treatments on NP models may be age-dependent [132]. For instance, the systemic administration of apocynin has exhibited age-dependent neuroprotective effects in a female mouse model of SCI by reducing excessive neuroinflammation through NOX-mediated ROS generation [133]. Since we only used young adult mice, studies with individuals of other ages are warranted.

Further studies are also necessary to confirm the putative sex-dependent effects of NOX2i on anxiety-like behavior and on the spinal cytokines profile, and to elucidate the mechanisms involved. The effects of other NOX2i doses, and treatment schemes, in which, for example, NOX2i is administered for the first time only after NP is fully established, should also be assessed.

Finally, it must be acknowledged that, despite its undeniable interest as a research tool, GSK2795039 itself cannot be considered to be a clinical drug candidate due to its poor pharmacokinetic profile [134]. Therefore, an eventual clinical translation would depend on the development of NOX inhibitors with full isoform-2 selectivity and improved pharmacokinetic profiles.

## 5. Conclusions

Our study demonstrates for the first time that short-term early systemic NOX2 inhibition with a small-molecule inhibitor can attenuate SNI-induced mechanical hypersensitivity, at least partly by reducing microglial activation at early timepoints. Furthermore, the influence of NOX2i treatment on the anxiety-like behavior and the spinal concentration of some cytokines appears to be sex-dependent in the SNI mouse model. Overall, our findings support selective targeting of NOX2 as a therapeutic strategy in NP, in both male and female individuals, with particular interest in females due to its apparent favorable impact on anxiety-like behavior.

## Figures and Tables

**Figure 1 biomedicines-11-00416-f001:**
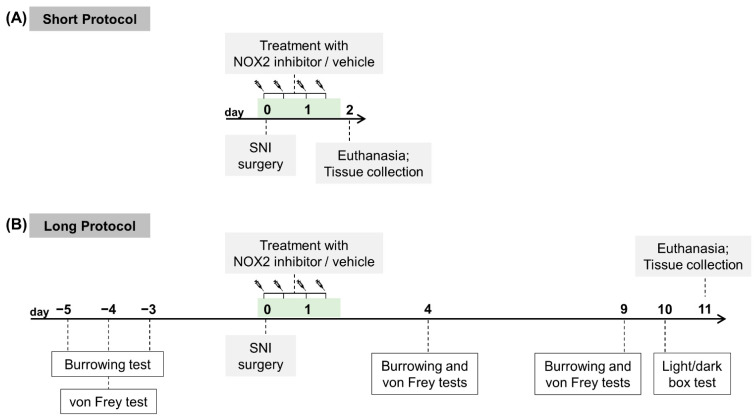
Schematic representation of the experimental design. Male and female mice were randomly allocated within sex to short (**A**) or long (**B**) protocol, and then to SNI-NOX2i or SNI-vehicle groups. NOX2 inhibitor (NOX2i) or vehicle solutions were subcutaneously administered, twice daily, for 2 days, starting 1 h before the spared nerve injury (SNI) surgery. SNI surgery day was set as day 0. Mice from the short protocol were euthanized on day 2, whereas mice from the long protocol were euthanized on day 11. Only the latter underwent behavioral testing.

**Figure 2 biomedicines-11-00416-f002:**
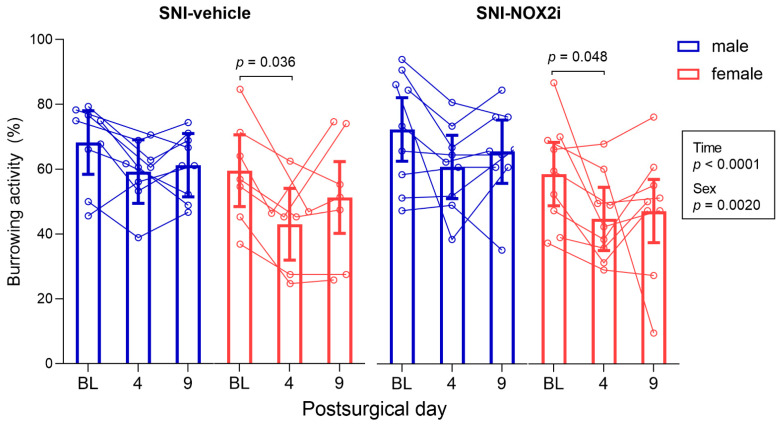
NOX2i treatment did not affect spared nerve injury (SNI)-induced reduction of burrowing behavior detected on day 4, particularly in female mice. The burrowing activity (%) was assessed before surgery (baseline, BL), and on days 4 and 9. Data were analyzed with repeated measures parametric analysis (mixed model), with factors *treatment*, *sex,* and *time*. Holm’s procedure was used to correct for multiple comparisons. Individual values and mean predicted values (95% CI) from the analysis are presented. *p*-values for statistically-significant main effects/interactions and planned comparisons are shown. ♂-SNI-NOX2i, n = 9; ♂-SNI-veh, n = 9; ♀-SNI-NOX2i, n = 9; ♀-SNI-veh, n = 7.

**Figure 3 biomedicines-11-00416-f003:**
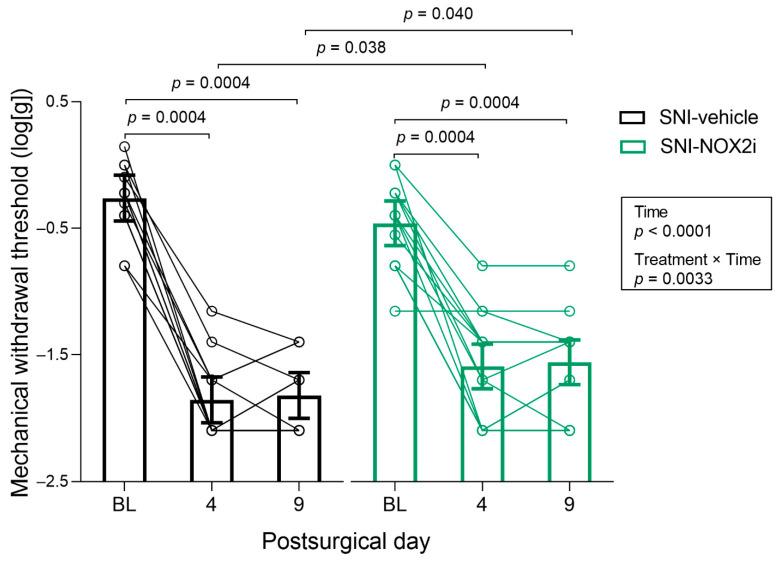
NOX2i treatment reduced spared nerve injury (SNI)-induced mechanical hypersensitivity in the ipsilateral hind paw. Mechanical withdrawal threshold (MWT) was assessed before surgery (baseline, BL), and on days 4 and 9. Data were analyzed with repeated measures parametric analysis (mixed model), with factors *treatment* and *time*. Holm’s procedure was used to correct for multiple comparisons. Individual log10-transformed values and mean predicted values (95% CI) from the analysis are presented. *p*-values for statistically-significant main effects/interactions and planned comparisons are shown. SNI-NOX2i, n = 14; SNI-veh, n = 13.

**Figure 4 biomedicines-11-00416-f004:**
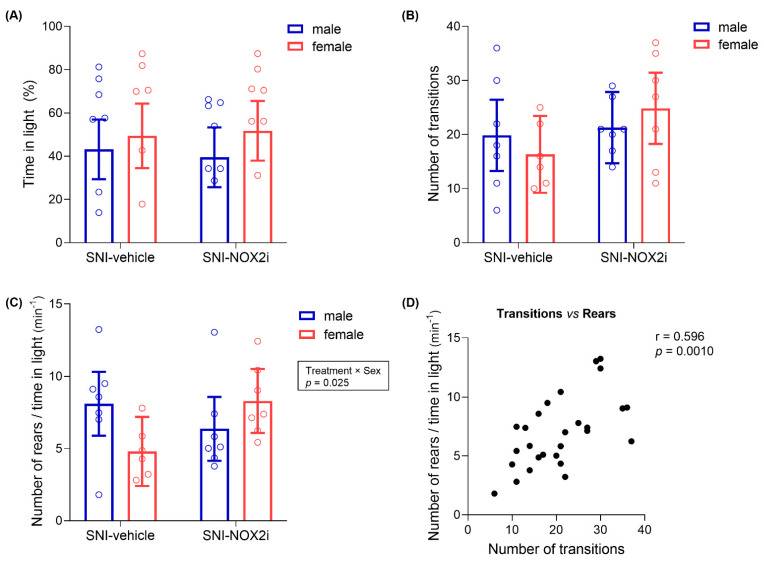
Effects of NOX2i treatment on the anxiety-like behavior after spared nerve injury (SNI), in the light/dark box (LDB) test, on day 10. NOX2i treatment affected neither the time spent (%) in the light chamber (**A**) nor the number of full-body transitions between light and dark chambers (**B**). The number of rearing events in the light chamber standardized by the duration of time spent in that chamber (**C**) was differently affected by NOX2i treatment in male and female mice. A positive correlation was found between the number of transitions and the standardized number of rears in the light chamber (**D**). For panels *A* to *C*, data were analyzed with single-measures ANOVA, with factors *treatment* and *sex*. Holm’s procedure was used to correct for multiple comparisons. Individual values and mean predicted values (95% CI) from the analysis are presented. *p*-values for statistically significant main effects/interactions and planned comparisons are shown. ♂-SNI-NOX2i, n = 7; ♂-SNI-veh, n = 7; ♀-SNI-NOX2i, n = 7; ♀ -SNI-veh, n = 6. For panel *D*, Pearson’s correlation coefficient analysis included all SNI animals (N = 27).

**Figure 5 biomedicines-11-00416-f005:**
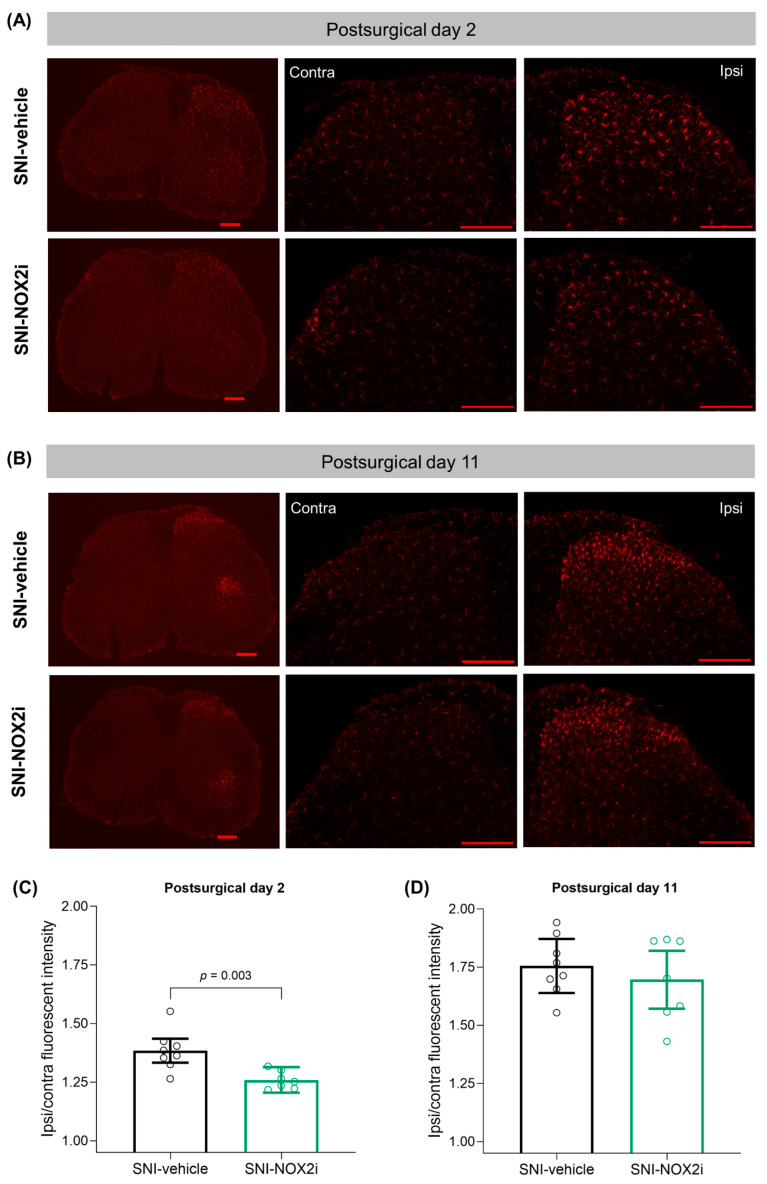
NOX2i treatment reduced spared nerve injury (SNI)-induced spinal microglial activation on day 2 but not on day 11. Representative images of the immunofluorescence staining with the microglial marker Iba-1 in the spinal cord of mice on days 2 (**A**) and 11 (**B**) are shown at 2.5× (left column) and 10× magnifications (middle and right columns; scale bar: 200 μm in all photos). Quantification of Iba-1 staining intensity in the dorsal horn on days 2 (**C**) and 11 (**D**) is represented as the ipsilateral/contralateral ratio of mean fluorescence intensity. Data were analyzed with single-measures ANOVA, with factor *treatment*. Holm’s procedure was used to correct for multiple comparisons. Individual values and mean predicted values (95% CI) from the analysis are presented. *p*-values for statistically-significant comparisons are indicated. Day 2: SNI-NOX2i, n = 7; SNI-veh, n = 8. Day 11: SNI-NOX2i, n = 7; SNI-veh, n = 8.

**Figure 6 biomedicines-11-00416-f006:**
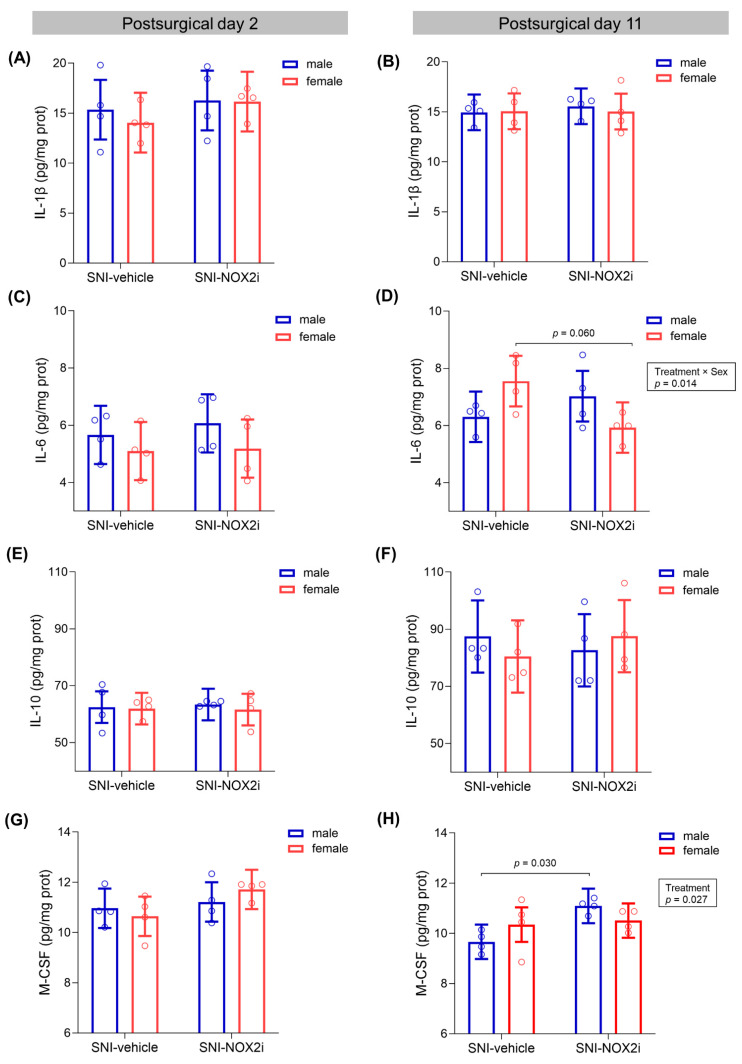
Effects of NOX2i treatment on spinal inflammatory cytokines after spared nerve injury (SNI) on days 2 and 11. On day 2, NOX2i treatment did not influence the spinal concentrations of interleukin (IL)-1β (**A**), IL-6 (**C**), IL-10 (**E**), and macrophage colony-stimulating factor (M-CSF) (**G**). On day 11, NOX2i treatment did not influence IL-1β (**B**) and IL-10 (**F**) concentrations, but differently affected the IL-6 concentration in males and females (**D**) and increased M-CSF values in males (**H**). Data were analyzed with a single-measures ANOVA, with factors *treatment* and *sex*. Holm’s procedure was used to correct for multiple comparisons. Individual values and mean predicted values (95% CI) from the analysis are presented. *p*-values for statistically- (or marginally-) significant main effects/interactions and planned comparisons are indicated. Day 2: ♂-SNI-NOX2i, n = 4; ♂-SNI-veh, n = 4; ♀-SNI-NOX2i, n = 4; ♀ -SNI-veh, n = 4. Day 11: ♂-SNI-NOX2i, n = 4; ♂-SNI-veh, n = 4; ♀-SNI-NOX2i, n = 4; ♀ -SNI-veh, n = 4.

**Figure 7 biomedicines-11-00416-f007:**
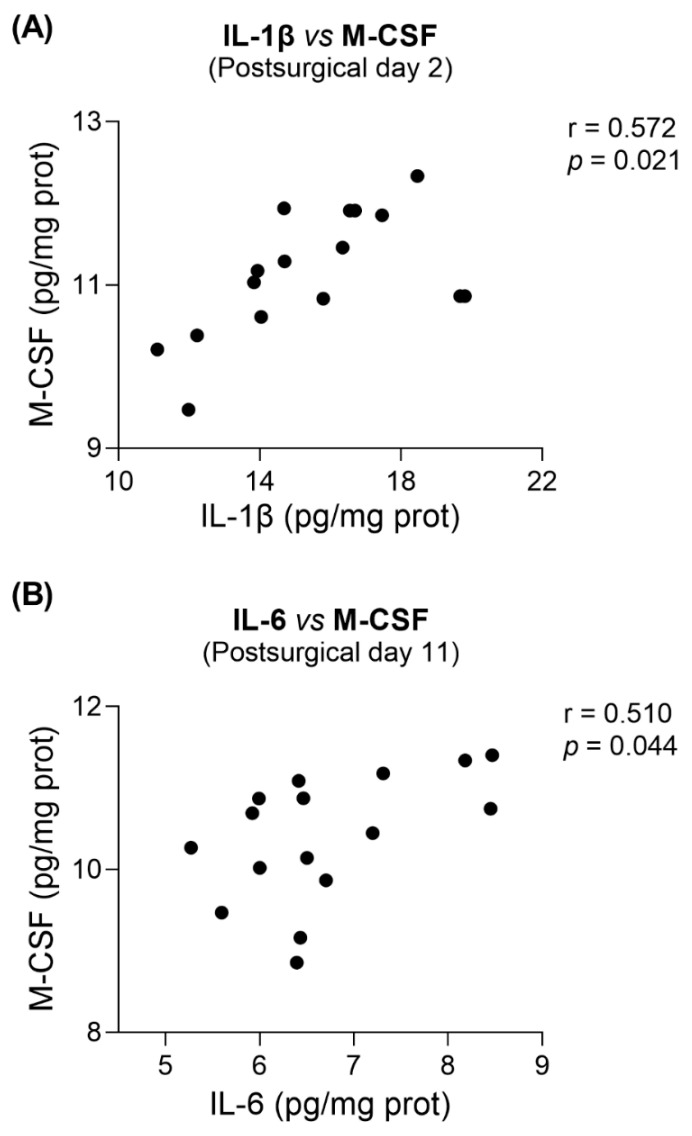
Positive correlations between IL-β and M-CSF were found at postsurgical day 2 (**A**), and between IL-6 and M-CSF at postsurgical day 11 (**B**). Pearson’s correlation coefficient analyses included all animals (Day 2: N = 16. Day 11: N = 16).

## Data Availability

The data presented in this study are available on request from the corresponding author.

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
