# Peer review of "Effects of NADPH Oxidase Isoform-2 (NOX2) Inhibition on Behavioral Responses and Neuroinflammation in a Mouse Model of Neuropathic Pain"

_biomedicines, 2023, doi:10.3390/biomedicines11020416_

Round 1
Reviewer 1 Report (Previous Reviewer 1)
I'm satisfied with the improvements done by the authors and suggest the acceptance of the actual state of the manuscript.
Reviewer 2 Report (Previous Reviewer 2)
It is ok
This manuscript is a resubmission of an earlier submission. The following is a list of the peer review reports and author responses from that submission.
Round 1
Reviewer 1 Report
The manuscript submitted by Teixeira-Santos et al. is well-written and interesting. However there are some points to be addressed:
- the authors should explain more clearly any dose-dependent effects of the NOX inhibitior used by their side. Several studies show that dose may interefere with the amplitude of the inflammatory response as well as may interefere with the viability of cells. So far, there is no clear-cut observation about the NOX inhibitor effects on this two issues!
- the authors are performing a quite challenging protocol with several procedures which may trigger some priming effects that may reverse the inflammatory response of microglia. The authors should discuss this issue and possible interference of trained immunity of microglial cells on the NOX expression and the inflammatory state. Please discuss the following papers: PMID: 29643512; PMID: 31781091
- the authors should also discuss whether any metabolic or maturation state of microglia (newborn vs adult or aged) may have an impact on the NOX2 behaviour responses. Some reports as listed here have shown the impact of maturation state and metabolism affecting the neuroinflmmatory status of microglia (PMID: 33806610; PMID: 33101271; PMID: 33239064);
- authors should consider to discuss broader the effects of NOX2 inhibitor in ROS production (please refer to this papers: PMID: 34685514; PMID: 25459294)
- the authors must report any signalling mechanism data for NFkB or other intermediate molecules!
- what about the role of MAPKs?
- the authors should provide more details in their immunohisto. images since they are not so clearly and the size is missing so far! Furthermore, they have to provide the original images as Supplementary File.
- the authors should keep in mind that for in vivo studies the IBA1 is not the ideal marker of microglial cells, since it may stain also other cells in the brain like pericytes (PMID: 24848101)! I suggest to comment this issue or to use another marker!
Please don't stickt only to my suggestins of papers given here! You may select also other relevant papers of your interest. My intention is only to give some advise what I find as very important to discuss here.
Reviewer 2 Report
The manuscript by Teixeira-Santos and colleagues, entitled "Effects of NADPH oxidase isoform-2 (NOX2) inhibition on behavioral responses and neuroinflammation in a mouse model of neuropathic pain" discusses selective inhibition of NOX2, through systemic treatment with GK2795039, on behavioral responses (pain and anxiety) and spinal neuroinflammation, suggesting that this approach may be a potential therapeutic strategy for neuropathic pain counteract. Additionally, authors in this study using male and female animals, seeing sex-dependent behavioral and biochemical differences, and increasingly emphasizing the importance of including gender in studies.
I appreciate the effort of the authors and I find their data interesting, and the English well written... but, there are some points that need to be addressed, because the work in this way, in my opinion, cannot be published.
· Materials and methods.
Although the authors have provided a supplementary image with the mice division, its reading is not easy. Animal division into groups should be included in the text of the main manuscript.
Line 84. "Eighty-six male and female C57BL / 6J mice" should become like "Eighty-six male (n=43) and female (n=43) C57BL/6J mice"
Line 101. "Sixty-four mice were randomly allocated within sex to one of two different experimental protocols" something like:
"the animals were randomized into the three treatment groups within the two experimental protocols:
- short protocol: sham (male n = 5, female = 5), SNI (male n = 8, female = 8) and SNI+inhibitor (male n = 8, female n = 8).
- long protocol: sham (male n = 4, female n = 4), SNI (male n = 9, female = 9) and SNI+inhibitor (male n = 9, female n = 9).”
In fact, the treatment with the inhibitor is carried out before inducing the pathology, so I guess, logically, that the division into treatment / experimental protocol was done simultaneously.
The text of lines 130-147 should be placed immediately below the division into groups.
Furthermore, from an a priori analysis you state that 8 animals per group is the size of your sample, but dummy animals are never 8 per group and have never been included in the statistical analysis of the data. This is incorrect.
Lines 84 and 149. “8 weeks-old upon 84 arrival “ and “NP was induced in 12 weeks-old mice by the SNI model”. So, between the arrival of the animals and the start of the experiments, 4 weeks pass. Why this choice?
2.4. Behavioral assessment. This paragraph should be better described, reporting the times and number of animals used. Something like this should be added:
"Behavioral assessments were conducted only on mice of long protocol. Baseline evaluation of burrowing (sham, 2 male and 2 female; SNI, 9 male and 7 female; SNI-inhinibitor, 9 male and 9 female) and mechanical allodynia (sham, 2 male and 2 female; SNI, 7 male and 6 female; SNI-inhinibitor, 7 male and 7 female) were performed in a three-day trial prior to SNI induction (see Fig. 1). Tests were repeated on the same animals at 4 and 9 days after SNI induction. The dark light box test was conducted only at day 10 after SNI (sham, 2 male and 2 female; SNI, 7 male and 6 female; SNI-inhinibitor , 7 male and 7 female).
Specify here why some animals were excluded from behavioral tests.
However, I find it unacceptable that on behavioral tests, only 2 control mice per sex were used. These data, to be published, must be brought to at least 4-5 animals per group and sex and included in the statistical analysis.
Figure S1 and 142. “Furthermore, an additional “extralong protocol” set was used 142 for the burrowing test and euthanized at a later timepoint.” “Animals from 1 of the 8 sets could not be submitted to von frey and dark test due to logistical constraints in the animal facility; the additional set of animals, submitted to an "extralong protocol" was only assessed for burrowing behaviour.”
The authors write that 1 set of animals was excluded from the study due to logistical problems, but they say that a second set of animals was excluded because it was subjected to an extra-long protocol ...
So I ask myself:
- it was not possible to submit to the extra-long protocol the animal set that did not carry out the tests due to logistical problems?
- why is there no ever mention of the extra-long protocol?
- why subject 1 animal per group to an extra-long protocol, statistically what relevance can it have?
- why if two sets of animals have been excluded (therefore 2 animals per group), does the number of SNI females go from 7 to 6, and not to 5?
2.4.1. Burrowing test. This test is simple to perform and low stress for the animal. That said, this test does not reflect pain (so remove any references to this). Although it is used in neuropathic pain models, and more generally in surgery models, it is more related to the stress/anxiety condition. Furthermore, it has been widely shown that this test varies greatly between the species under study. Authors use nesting material, but the literature reports that the use of pellets is preferable for C57BL/6J mice.
Also, the text of lines 313-325 should be summarized and incorporated into the materials and methods, not the results. Animals excluded from the test, which if I am not mistaken is only one animal (female SNI), must be reported here.
2.4.2. von Frey test. Authors are requested to indicate all 5 forces used for von Frey filaments and to indicate the manufacturer. Furthermore, it should be reported how the authors arrived at the final value reported in the graph, it is not clear.
2.4.3. Light/dark box test. Line 223. “These parameters, which have been considered as a measure of anxiety in this apparatus”.
As written, it is incorrect. Indeed, a decrease in the number of transitions and rearing events are considered an anxious-like behavior. Although, rearing events are not a reliable data. I suggest to the authors, since they have the recordings, to integrate the latency time, which is a more reliable measure.
Where the latency corresponds to the time in which the animal first passes from the white compartment (where the animal has been placed) to the dark one. A decrease in time corresponds to anxious-like behavior.
2.5. Tissue collection. Also in this paragraph the exact number of animals used should be reported.
Statistical analysis and results. Statistical analysis and interpretation of the data is not appropriate. Authors should re-form all their findings and statistical analysis.
- as already mentioned sham animals must be brought at least 4-5 per group and sex and included in the statistical analysis.
- why are there no sham treated with inhibitor (GSK ..)?
- in the legend of each figure, the statistical analysis used must be reported.
- the significant differences (all) must be reported in the graph and not only in the test (either as p = XXX or as a symbol ***).
- if the authors want to represent the data as columns (mean) and dots (value of single animal), they should use a graph capable of representing all the mice in the study (even if two values ​​are identical, there should be 2 dots; see von frey: there are 14 animals/group, but they are 3 dots).
- the shams should be represented like the other groups, so columns and dots (not just dots).
- in the tests where the variables are 3 (burrowing and dark/light), the most appropriate analysis is a three-way ANOVA (time, treatment and sex). Two-way ANOVA (2 variables, time and treatment) should be used for the statistical analysis of the MWT. Also for biochemical analysis, two-way ANOVA (2 variables, treatment and sex) should be used. For all the most appropriate post-hoc test is a Bonferroni.
Lines 391-393: "Nevertheless, vehicle-treated SNI-females showed a more distinctive behavior, characterized by a lower number of both transitions between the two compartments and rearing events than sham-operated individuals".
How do the authors tell if, sham animals were not included in the statistical analysis?
Immunofluorescence. How was the fluorescence intensity calculated? why have the cells not been quantified? I don't understand why the authors did iba1 as immune and then cytokines as ELISA. Couldn't everything be done as immune and also see colocalization between microglia-cytokines? Why weren't ROS evaluated?
Other. Authors attempted to assess the anxious condition, but the number of animals is too small. In mood behaviors you work with many more animals. So,
- why was anxiety evaluated and not depression, which is usually more associated with pain, with a strong implication of ROS?
- because having done some mood tests, the neuroinflammation and the effect of the treatment in important brain areas such as hypothalamus, hippocampus, amygdala, thalamus or prefrontal cortex have not been evaluated?
